# A New Approach Using AHP to Generate Landslide Susceptibility Maps in the Chen-Yu-Lan Watershed, Taiwan

**DOI:** 10.3390/s19030505

**Published:** 2019-01-26

**Authors:** Thi To Ngan Nguyen, Cheng-Chien Liu

**Affiliations:** 1Faculty of Geology, University of Science, VNU-HCM, Ho Chi Minh City 700000, Vietnam; 2Department of Earth Sciences, National Cheng Kung University, Tainan 701, Taiwan; ccliu88@mail.ncku.edu.tw

**Keywords:** landslide susceptibility map, correlation statistics, AHP, Chen-Yu-Lan watershed, disasters

## Abstract

This paper proposes a new approach of using the analytic hierarchy process (AHP), in which the AHP was combined with bivariate analysis and correlation statistics to evaluate the importance of the pairwise comparison. Instead of summarizing expert experience statistics to establish a scale, we then analyze the correlation between the properties of the related factors with the actual landslide data in the study area. In addition, correlation and dependence statistics are also used to analyze correlation coefficients of preparatory factors. The product of this research is a landslide susceptibility map (LSM) generated by five factors (slope, aspect, drainage density, lithology, and land-use) and pre-event landslides (Typhoon Kalmaegi events), and then validated by post-event landslides and new landslides occurring in during the events (Typhoon Kalmaegi and Typhoon Morakot). Validating the results by the binary classification method showed that the model has reasonable accuracy, such as 81.22% accurate interpretation for post-event landslides (Typhoon Kalmaegi), and 70.71% exact predictions for new landslides occurring during Typhoon Kalmaegi.

## 1. Introduction

Landslides are disasters which often occur in hilly or mountainous places in the rainy season. They are classified as a category of disasters triggered by hazards related to geological processes (earthquakes, volcanic eruptions, floods in mountainous areas) and meteorology (heavy rains, storms, or typhoons) [1,2]. According to a statistical report on disasters in the last decade (2006–2015) produced by the International Federation of Red Cross (IFRC), landslides account for 4.7% (177 events) of the total number of reported natural hazards [1]. The highest density of landslide occurrences is in Asia, accounting for 118 of the 177 reported landslides occurring worldwide [1]. A report by the The World Bank, “Natural Disaster Hotpots: A Global Risk Analysis” (2005), noted that Taiwan is a small country with an area of approximately 36,000 km^2^ and a population of about 23 million people, but is classified as a high-risk country for such disasters [3]. Taiwan ranked first of the top 15 countries (based on land area) with regard to being the most exposed to multiple hazards (affected by three or more disasters) [3]. In other statistics examining countries which are at relatively high mortality risk from multiple hazards, Taiwan also ranked first of the top 35, with 95.1% of its population living in areas at risk [3]. Besides earthquakes and typhoons, landslides are the most common natural disasters in Taiwan. Landslide inventory reports of The GEODAC (Global Earth Observation and Data Analysis Center) show that the total landslide area in Taiwan increased from 169.93 km^2^ in 2004 to 392.44 km^2^ in 2015, and the area was highest in 2010 (reaching 506.09 km^2^), corresponding to the post-Typhoon Morakot period in August 2009. Most landslides here are triggered by typhoons and heavy rains, and sometimes by large earthquakes.

M. J. Crozier [4] proposed that the factors influencing the landslide occurrence could be classified into two main categories, preparatory and triggering. The triggering group is external factors that cause immediate changes in the strength/stress of the slopes, resulting in movement, such as earthquakes, volcanic eruptions, typhoons, and heavy rainfalls, whereas the preparatory group is potential factors, such as slope failures, material composition, and coverage, which cause sliding due to gravity [5,6,7,8]. Landslides are one of the three major geological hazards (earthquakes, landslides, subsidence) occurring in Taiwan, and typhoons and earthquakes are the two main triggering factors. However, more than 10 typhoons hit Taiwan annually, and most landslides are caused by typhoons. Understanding the relationship between the rainfall thresholds that can trigger landslides and landslide susceptibility is very significant to enable better warnings of landslides during the typhoon and rainy season. Therefore, the research project is divided into two phases as follows: phase one is to produce the landslide susceptibility map using potential or preparatory factors, phase two is to find the correlation between the precipitation thresholds that can trigger landslide occurrences and the landslide susceptibility index (also called the relationship between triggering and preparatory factors). This article describes phase one of the research project.

There are many studies of landslide susceptibility modeling that have been developed in Taiwan and around the world using different approaches, such as statistical (multivariate statistics, bivariate statistics) [7,8,9,10,11,12,13], probabilistic [14], deterministic [15], and even heuristic approaches [16,17,18,19]. These studies focus on developing models for calculating the landslide susceptibility index using various parameters of preparatory factors (slopes, lithology, land-use, and so on). Besides, some other studies also use a rainfall parameter in their models [20,21]. Generating a landslide susceptibility map by using the AHP method has been carried out by many researchers worldwide [16,17,18,19,22,23]. Most of these studies applied expert opinions for dividing the classes of each parameter (factor) and for setting the values of pair-wise comparisons (the pair of classes or parameters). The AHP helps decision-makers find the best choice and thus better understand problems. Based on mathematics and psychology, AHP was developed by Saaty in the 1970s and has been expanded and supplemented ever since [24]. AHP provides an accurate framework for structuring a problem that needs to be resolved. It has a special application in group decision-making, and is used around the world in many critical contexts, such as government, business, industry, healthcare, education, and even disaster planning [19,25,26,27,28,29]. Although commonly used, the AHP often has limitations because of the inability to incorporate uncertainties and inherent inaccuracies in the data, which are related to the mapping between the perceptions and judgment of the decision maker and the precise numbers used in the AHP model. After decomposing an unstructured situation into small parts and building a hierarchy, it is important to establish pairwise comparative analysis of the importance of various indicators in decision making. In most previous AHP studies, the authors made these pairwise comparisons based on a synthesis of expert opinions or experiences [19,23,25,26,27,28]. However, in this study, we assign numerical values for subjective comparisons of the importance of pairwise comparisons by comparing the correlation between landslide area and distribution area of elements, using bivariate analysis and correlation analysis statistics. On the other hand, the different areas have different characteristics of landslide factors, and hence the probability of landslides is not the same. In some cases, the landslide occurrences did not follow the theoretical rules, such as the highest slope angle did not have the highest probability of landslides, or an area with lower drainage density had lower landslide probability. Beside the slope angle, the probability of landslides is also influenced by the ground material’s hardness or coverage, and so on. Therefore, it is not objective just to rely on expert opinions to set the values of pair-wise comparisons. To solve this problem, historical landslide data was used to calculate the values of pair-wise comparisons (classes or parameters) using bivariate statistics and correlation statistics. This is a new approach to apply the AHP in the calculating weights of landslide factors.

M. T. Suzen et al. carried out a survey of 145 studies on the modeling of landslide susceptibility (from 1986 to 2007), and found that four major groups of parameters were used to analyze the landslide susceptibility index, including geological parameters (geology/lithology, fault/lineaments, strata-slope interactions), topographical parameters (drainage, surface roughness, elevation, slope angle, aspect, curvature, slope length) geotechnical parameters (soil thickness, soil texture), and environmental parameters (land-use/land-cover, anthropogenic parameters, rainfall, positions within catchment) [30]. Of these, the five most commonly used parameters were slope angle (97.9%), geology/lithology (92.31%), land-use/land-cover (75.52%), drainage (72.73%), and aspect (59.44%) [30]. This suggests that the above five categories are important factors for landslides. In addition, N. Casagli et al. also suggested that the three key factors causing landslides were topography (slope failure), material composition (geology/lithology), and land-use [2]. Based on the above analysis, five parameters, including lithology, slope angle, aspect, drainage density and land-use, were used to analyze the landslide susceptibility index in this study.

### Study Area

The Chenyulan river watershed is located in the central part of Taiwan, in the island’s major fault zone (slate belt) (Figure 1). This river is a tributary of the Zhoushui river. There are important fault systems in this area, such as Chenyulan, Dili, Kuling Chiao, Shenmu, and so on. The average elevation is 1680 m above sea level, and the average slope is 34.5°. The erosion activities here are complex, and occur at high densities. The area of landslides is 16,598,445 m^2^, or 4.2% of the total of 392,444,312 m^2^ in Taiwan. This area is also affected by the tropical monsoon climate of Taiwan, which is warm and humid all year. The rainy season is from May to October, and typhoons often occur in the summer and autumn, from June to October, more frequently in August and September, with on average four direct hits per year [31]. The annual precipitation is more than 2500 mm. The rainfall in Taiwan often concentrates in mountainous areas, and therefore the Chen-Yu-Lan river watershed is also one of the areas with the highest rainfall in Taiwan. Summer rainfall accounts for around 80% of the annual precipitation.

The study area’s lithology can be divided into the following: metamorphic rocks are distributed in the east, sedimentary rocks are located in the west, terrace deposits (gravel, sand, clay) are scattered from the central to the south, and alluvium is concentrated along the main river in the north. Geological structures here are very complicated, with many faults and fold systems which are distributed from the north to the south of the area. The fault systems are in two main directions, including the north-south (Chenyulanchi fault, Shanshilchia fault, Dili fault, and Kulingchiao fault) and the east-west (Shenmu fault, Shihpachekeng fault, and Erhyu fault). The fold systems are arranged in the directions of the north–south (Pinglin syncline) and the north northeast–south southwest (Hoshe anticline and Tungfushan syncline). The Chen-Yu-Lan River is located on a large fault (Chenyulanchi), where a recent significant earthquake (Chi-Chi) occurred in 1999. For all these reasons, this area is one of the most vulnerable to landslides in Taiwan.

Based on a statistical report of the IFRC, the highest number of landslides occurred in the period from 2008 to 2010 (accounting for 72 of the 177 events occurring worldwide) [1]. This period also corresponds to some very strong typhoons in Taiwan (Ex: Typhoon Kalmaegi 2008, Typhoon Morakot in 2009). As typhoons that landed on the main island are often accompanied by enormous amounts of rainfall, they often trigger shallow landslides, such as debris flows or slides, and these are two common categories of landslides in the Chen-Yu-Lan river watershed. Typhoon Kalmaegi, which formed on 13 July, brought heavy rainfall of approximately 1200 mm over a three-day period (16 July to 18 July 2009), caused a loss of lives (25 fatalities and one missing) and destruction of about 332.3 million USD of property [31]. Typhoon Morakot, which formed on 2 August and dissipated on 12 August 2009, hitting Taiwan from the 8 August to 9 August, brought much heavier rainfall of 3000 mm caused a greater loss of lives (677 deaths, four severely injured, 22 missing, and 25 unidentified bodies) and damage of about 274 million USD. Based on sustained winds ranging from 118 to 156 km/h (Typhoon Kalmaegi: 120 km/h; Typhoon Morakot: 140 km/h), the Regional Specialized Meteorological Center Tokyo (RSMC Tokyo) ranked Typhoon Kalmaegi and Typhoon Morakot in the typhoon group of the Tropical Cyclone Intensity Scale. However, the impacts (damages and fatalities) of these two typhoons in Taiwan are very different (as mentioned above) because of the huge difference in their rainfall (Typhoon Kalmaegi: 1200 mm; Typhoon Morakot: approximate 3000 mm). Due to this difference, in this study, we considered Typhoon Kalmaegi as a normal event and Typhoon Morakot is an extreme event which triggered landslides. Landslide data related to these events are thus used to generate a landslide susceptibility map and evaluate the accuracy of the predictions of our combined model.

## 2. Materials and Methods

Figure 2 shows the flowchart of this research, the types of data and the steps to conduct the study are as follows:

### 2.1. Producing Landslide Inventory Maps Related to Events

Unclouded Formosat-2 satellite images taken from 10 June 2008 to 22 June 2008 (the period before Typhoon Kalmaegi occurred), from 24 August 2008 to 3 February 2009 (the period after Typhoon Kalmaegi), and from 18 October 2009 to 21 October 2009 (the period after Typhoon Morakot) were used to produce landslide inventory maps (LMs) related to the event (Typhoon Kalmaegi), which are called, in turn, the pre-event LM and the post-event LM, using the Formosat-2 Automatic Image Processing System (F-2 AIPS) [32,33,34]. The product of this processing is called a spectral summation intensity modulation (SSIM) image. The SSIM images thus obtained were combined with the digital elevation model (DEM) data to produce 2D (Figure 3a) and 3D (Figure 3b) composite images [32]. The DEM is presented as a raster graphics image with a resolution of 20 m, and is able to show the height information of the surface of the study area, which was used to assist in the preparation of the landslide inventory [34] (Figure 3a,b).

The Expert Landslide and Shaded Area Delineation System (ELSADS) was then used, along with the differentiation thresholds of the Normalized Difference Vegetation Index (NDVI) and the normalized green red difference indices (NGRDI) to automatically detect and localize vegetation coverage and non-coverage areas, as shown by the green lines in Figure 3a,b [34,35]. The resulting flexible, composite 3D images (Figure 3b) can enable users to easily recognize and re-detect actual landslides, as well as erase buildings, roads, rivers, farms (vegetation non-coverage areas) and so on to detect the real landslides, as shown by the yellow lines in Figure 3a,b. This work was used to produce event-based landslide inventory maps in the whole study area (Figure 4). Most of the landslides in the Chen-Yu-Lan watershed are shallow ones that are known as debris flows and debris slides. Overlapping two kinds of LMs (the pre-event LM and the post-event LM) generated the during-event LM (this LM showed new landslide positions which occurred during the event) (Figure 4).

### 2.2. Building a Hierarchy Tree of Landslide Factors

To build a hierarchical tree, factors (or variables) were classified into components (classes) based on experts’ experience. The DEM in 20 m resolution (Figure 5a) is supported by the Forestry Bureau, Council of Agriculture in Taiwan was used to extract slopes (SL), drainage densities (DD), and aspects (AS) (Figure 5d–f). Besides, a geological map (GM) (scale 1:250,000 was used to measure the strength of the material. Based on the information about the rocks’ resistances and distributions, the lithology in this area was divided into six main groups, namely deposits, alluvium, sandstone-shale, argillite-slate, quartzite slate, and phillite-slate (Figure 5b). This GM data was then converted to the raster data in 20 m resolution. A land-use map (LU), which was published by Council of Agriculture and Central Geological Survey in 2006, was used to determine the coverage of the surfaces. This data was processed by SPOT 5 satellite images which were taken in 2004. The LU is classified into six major groups, including agriculture, forest, road and building, river body and wetland, grassland, and bare land (Figure 5c). The LU data was also converted to the raster data in 20-m resolution.

### 2.3. Computing the Landslide Density of Factors

Instead of based on expert opinion analyses to assess the important level of class pairs in each factor, we used the landslide density of each class to compare and calculate this important level. The landslide density of class *i* (*D_i_*) is calculated in Equation (1) with the pre-event LM data, is ratio of landslide area in class *i* (*L_i_*, pixel unit) and distribution area in class *i* (*A_i_*, pixel unit).
(1)$$D_{i}=\frac{L_{i}}{A_{i}}$$

### 2.4. Calculating the Weight of Factors by Combining of AHP and Bivariate Analysis

Then, this comparison is made between pairs of indicators (classes) and is combined into a matrix of *n* lines and *n* columns (*n* is the number of indicators or classes in each factor) (Figure 6). The element *D_ij_* represents the important level of the index *i* (line) versus the index *j* (column). The relative importance of indicator *i* versus *j* is calculated in *D_i_*/*D_j_* ratio, and vice versa of indicator *j* versus *i* is *D_j_/D_i_*, and *D_ii_* = 1 (Figure 6). Each matrix has been set up for each factor that can set the priorities of the elements (classes) in the hierarchy tree. The priority is a nonnegative number that fluctuates from 0 to 1 representing the weight association in each element at each level. By definition, the priority of the target is 1, and the total priority of a level is also 1. In this study, the priority is also the weight (*W*) of each class, and computed in Equation (2), where n is number of classes in each factor, *D_i_* is the landslide density of class *i* (Equation (1)), and *W_i_* is the weight of class *i*.

(2)$$W_{i}=\frac{D_{i}}{\sum_{i=1}^{n}D_{i}}$$

After executing steps of calculating the weights of classes, one question was set forth whether any method evaluates the validity of the significance values of the indicators (classes). According to Saaty [36], the consistency ratio (*CR*) can be used and calculated in Equations (3)–(5), where *CI* is the consistent index, *RI* is the random number, and n is number of classes in each factor. This ratio compares the consistency with the random objectivity of the data. The consistency ratio (*CR*) of less than 0.1 is acceptable. If it is greater than 0.1, the decision maker needs to reduce the heterogeneity by changing the significance level between the pair of indicators (classes).
(3)$$\lambda_{\max}=\sum \sum_{i=1}^{n}D_{i}\sum_{j=1}^{n}D_{j}$$
(4)$$CI=\frac{\lambda_{\max}-n}{n-1}$$
(5)$$CR=\frac{CI}{RI}$$
For each of the *n*-level comparison matrices, Saaty [36] tested the creation of random matrices and calculated the *RI* (random index) corresponding to matrix levels as Table 1.

### 2.5. Finding Out Coefficients of Landslide Factors by a Combination of AHP and Correlation Model

In order to measure and compare the importance or influence level of pairs of factors on landslides, the correlation and dependency model is used. The weighting correlation of landslide factors is analyzed on each pixel, then aggregated and output, and is called the correlation (*r*). According to Pearce, correlation (*r*) is a statistical indicator that measures correlation between two variables (*x* and *y*) [37]. The correlation (*r*) fluctuates from (−1) to 1. If the *r* equals or approximately 0, two variables have no relation, and vice versa if the *r* is −1 or 1, two variables have an absolute relation. The process of calculating correlation coefficient in turn is performed in the order of steps as explained in formulas 6 to 10.

Firstly, the average weight value of factor *k* (*µ_k_* ) must be calculated throughout the study area, as in Equation (6), where *W_ik_* is the weight value of pixel *i* of factor *k*, and *n* is number of pixels in the study area. Then the variance (*Var_k_*) and standard deviation (*σ_k_*) of factor *k* are calculated, as in Equations (7) and (8). The covariance (*Co*var*_kj_*) between the weight values of factors *k* and *j* is calculated as in Formula (9). The variance and covariance of factors are represented by a variance—covariance matrix, and shown in a symmetric matrix (Table 2).
(6)$$\mu_{k}=\frac{1}{n}\sum_{i=1}^{n}W_{ik}$$
(7)$$Var_{k}=\frac{\sum_{i=1}^{n}{\left(W_{ik}-\mu_{k}\right)}^{2}}{n-1}$$
(8)$$\sigma_{k}=\sqrt{\frac{\sum_{i=1}^{n}{\left(W_{ik}-\mu_{2}\right)}^{2}}{n-1}}$$
(9)$$Co{\mathrm{var}}_{kj}=\frac{\sum_{i=1}^{n}\left(W_{ik}-\mu_{k}\right)\left(W_{ij}-\mu_{j}\right)}{n-1}$$

The correlation (*r_kj_*) between the weighting values of the pair of factors (*k* & *j*) is calculated by the Formula (11), where *σ_k_* is the standard deviation of factor *k* and *σ_j_* is the standard deviation of factor *j*. These correlations are given the absolute value also combined into a symmetric matrix (Table 3) which is applied for calculating coefficients (*C*) by the AHP model (Section 2.3).
(10)$$r_{kj}=\frac{Co{\mathrm{var}}_{kj}}{\sigma_{k}\sigma_{j}}$$

### 2.6. Calculating LSI by a Combination of Coefficients and the Geometric Multivariate Model

Incorporate the factors correlation coefficients with their weights in Equation (11) to calculate the LSI, where *W* is the weight of factors, *C* is the coefficient of factors, and *n* is the number of factors.
(11)$$\mathrm{LSI}=\prod_{i=1}^{n}W_{i}^{C_{i}}$$

## 3. Results and Discussions

### 3.1. The Weight of Landslide Factors Calculated by the Combination of AHP and Bivariate Analysis Model

Table 4 shows a logical correspondence between the weight calculated by the AHP method and the properties of the classes (indicators) of the factors such as SL, GM, LU, and AS. For the SL factor, the weight increases proportionally with the gradient of the slope. Although the distribution areas of the SL classes are uneven (the largest is approximately 50% and the smallest is 2%), this also does not govern the reliability of their weight calculations. For example, the weight (0.580) of the greater than 100% class (distribution area 17%) is much larger than the one (0.234) of the 55–100% class (distribution area 53%). This shows the importance of SL factor for landslide occurrences. The LU case is the same as the SL one. Although the forest cover is over 70% of the study area, its weight (0.034) is less than the bare-land’s (0.657) (the distribution area is approximately 5.4%).

Low-resistance petrographic groups due to being compressed and crumbled by tectonic activities such as phyllite-slate-interbedded sandstone or quartzite-slate-coaly shale correspond to high and the highest weights, whereas high-strength rocks formed in less severe environments such as sandstone, shale, and argillite are low in weight. Although deposits and alluvium are of low-strength materials, their weights are not high. This inadequacy can be explained by the fact that these types of materials are distributed in areas with flat terrain and low gradient.

From the results of calculating the weights of the aspect factors, it is found that the south and west landslide risk is higher than the other directions. The distribution of the aspects in the study area is fairly uniform (in the 10–17% range), so the difference in weight of these classes is not as great as in the case of slopes, land use. The case of DD is similar to that of aspects that the difference in weight values of classes is not large. It is also unreasonable for smaller DD values (0.002–0.003) to have greater weights in a comparison to larger one (0.004–0.005) (Table 2). This can be explained by two reasons: the influence of the distribution area on the weight calculation of DD classes and almost DDs 0.002–0.003 are distributed in locations with steep slopes.

The result of the computation of the consistency ratio shows that all *CR*s less than 0.1 (Table 5), which mean that the values of significance between pairs of indicators are acceptable.

### 3.2. Calculating Coefficients of Landslide Factors by the Combination of AHP and Correlation Model

Table 6 depicts the symmetric matrix of landslide factors’ coefficients that calculated by the AHP. This shows that GM and LU are two important factors in computing LSI because their coefficients are higher than the others’. The two factors with the best correlation are LU and GM, whereas DD has a poor correlation with the other factors.

### 3.3. Generating the LSM by the Combination Model of the Geometric Multivariate and Factor Coefficients

A landslide susceptibility map (LSM) was produced by as in the Equation (11), where the significance of landslide factors is also the factor’s coefficient (*C*) (Figure 7a). The LSM showed that most of landslide areas (black color) which occurred in events (Typhoon Kalmaegi and Typhoon Morakot) distributed in the highest indices (red color) (Figure 7b–e). The safe regions are distributed in low-index positions corresponding to the river channel and its neighborhood with fairly flat terrain and low gradient slopes. Figure 8 shows how to determine the thresholds of the LSM indices. Two cumulative percentage curves of landslide and non-landslide are set and viewed in the same graph, the intersection of which is the threshold (the dotted red line) between the safe zone (accounting for more than 80% of study area) and the dangerous zone (approximate 20% of study area) (Figure 8a). Areas in the danger zone are highly coincident with real landslides. After comparing and reviewing the landslide data of different years, we found that the landslides that occurred afterwards (new landslides) often occurred in adjacent locations along the old landslides. Precise warnings about areas of landslide are very significant, therefore, in order to increase the accuracy in warning, we create additional buffer zones for warning areas and then two sub-thresholds are set up (Figure 8b). Figure 8b shows the distribution percentage of LSI corresponding to actual landslides and non-landslides. This figure shows that the landslide curve is variable (at the threshold LSI (0.1458), the landslide percentage increases suddenly from less than 10% to greater than 60%), while non-landslide does not fluctuate (Figure 8b). Therefore, based on the variation of landslide curve (Figure 8b), two sub-thresholds are also defined (the dotted brown and green lines). As a result, the LSM indices were divided into four levels, including low (index: 0.0148–0.0927), low medium (0.0928–0.1161), high (0.1162–0.1457), and very high (LSI: 0.1458–0.4123), with distribution areas of 30.25%, 30.52%, 19.92%, and 19.31%, respectively (Figure 8 and Figure 9).

Compared to the other factors in the LSI modeling calculations, the GM here does not have a high resolution (scale: 1:200,000), so the rock classification was not detailed, which are grouped together for example the argillite-slate-phyllite group (three rock categories were grouped together). In addition, because GM’s landslide coefficient (or significant level for landslides) is higher than other factors’ ones, high LSI indices are roughly aligned with the geological boundaries of the GM classes.

As result, the very high LSI (0.39–0.41), which occupies about 46.11 ha, is mainly distributed in the southern part of the study area (Yu-Shan mountain peak) (Figure 7a). Although the LSI value is high, there are some positions that do not have landslide occurrences (non-landslides). This can be explained by the fact that the LSI values of these positions are dominated by three factors including slope, lithology and land-use. All of these factors have the highest weighting in above positions, thus making the LSI values very high. Although these positions are located in steep cliffs and are in the bare-land group, the exposed surface is bedrock rather than loose materials, so landslides do not occur here.

### 3.4. Validating the LSM with Types of Related-Typhoon Kalmaegi LMs

The binary classification and Kappa index (*K*) methods were used to validate the LSM. The “binary classification” method is a method of assessing the accuracy of interpretative models, which is widely used in research topics related to statistics in the fields of economics, medicine, or GIS [7]. The binary classification is a model, in which objects were classified into two groups based on its attributes. In this research, the attributes were classified in particular are landslide occurrence. Therefore, the LSI is divided into two groups consisting of dangerous and safety groups. A threshold of division between danger and safety is established.

Here, the correct interpretation is called accuracy, which is the sum of *TP* and *TN*. *TP* is the area of real landslides corresponding to LSI area that is divided into the dangerous group. *TN* is non-landslide areas and corresponds to LSIs that are partitioned in safe areas. In contrast, the error is the sum of *FN* and *FP*, where *FN* is the area to be interpreted as having a safe LSI (safe group) but with landslide occurrences, and *FP* is the area to be interpreted as a high risk of landslides (dangerous group) but no landslide occurs. The study area is a total of *TP*, *TN*, *FP*, and *FN* (Equation (13)).

The Kappa index method developed by Cohen Kappa (1960) is used to evaluate the similarity between the statements. Specifically, in this study, there was a claim of landslide or no landslide. According to Cohen Kappa, the main interpretations of *K* can be divided into the following: (*K* < 0.2): poor agreement, (0.2 <= *K* < 0.4): fair, (0.4 <= *K* < 0.6): moderate, (0.6 <= *K* < 0.8): good, and (0.8 <= *K* <= 1): very good consensus. The *K* index is calculated in Equations (13)–(16), where *P*_0_ is the ratio between the interpretation and the actual situation, and *P_e_* is the desired proportion.
(13)Total=TP+TN+FP+FN
(14)$$P_{0}=\frac{TP+TN}{Total}$$
(15)$$P_{e}=\frac{\left(TP+FP)(TP+FN)+(TN+FN)(TN+FP\right)}{{\left(Total\right)}^{2}}$$
(16)$$K=\frac{P_{0}-P_{e}}{1-P_{e}}$$

Table 7 showed that the prediction accuracy of the LSM is high if overlapping to the normal events (Typhoon Kalmaegi: the post-event 77.35%, the during-event 70.26%; Typhoon Morakot: the post-event 65.64%, the during-event 58.49%). The calculating *K* index results are also the consensus of predictions of Typhoon Kalmaegi (post-event: *K* = 0.559; during-event: *K* = 0.412) was higher than Typhoon Morakot (post-event: *K* = 0.313, during-event: *K* = 0.170). These *K* values are at moderate consensus. This can be explained by the fact that the rainfall of these two events is very different. Kalmaegi is classified as a normal event (maximum total of precipitation: 774 mm), while Morakot (maximum total of precipitation: 1974 mm) is classified as an extreme event. To increase interpreting agreement, a rainfall factor should be considered to add into the computation model. Most of the previous studies [21,38], researchers have often considered rainfall as one of the landslide factors and they add it to the other LSI computational model. However, in our project, finding thresholds between the rainfall and the LSI indices is very significant in the warning of landslides before the storm occurs. Therefore, the rainfall is considered as one of the trigger factors, so this parameter is not included in this computational model (Phase 1), it will be considered in phase 2. In the upcoming work, the relationship between the rainfall and LSI indices will be analyzed from which to find their corresponding thresholds (Phase 2). This work is thus done in phase 2.

Besides, success rate curves have been used in many studies to assess the validity of predictions produced under different scenarios, and they have also been applied in some works that obtained LSI maps using different factors [39,40,41,42]. The “success rate curve” is a method which has been used to measure the accurate proportion of prediction by plotting cumulative percentages of high LSI risk and real landslide during typhoon. Therein, the data of overlapping LSI values and actual landslides were analyzed, and based on the resulting statistics, the cumulative percentages of LSI and landslide were calculated. These results were then plotted on the same graph, known as a “success rate curve”. By observing the distributions of the curves in the graph, it is possible to assess the accuracy of the related predictions.

The LSM’s success rate curves which are evaluated by events (Typhoon Kalmaegi and Typhoon Morakot) showed a high accuracy of interpretation rates with approximately 86% of the real landslides (post Typhoon Kalmaegi) occurring in areas with high and very high LSI (dangerous distribution area: 39%), and over 75% of new landslides occurring during this event. Also, 61.73% of actual landslide areas occurred post Typhoon Morakot and 49.37% of new landslide positions also occurred during the typhoon (Figure 9, Table 8). These curves once again show the importance of rainfall parameters in the LSI calculation model for extreme events in the comparison of the results between Typhoon Kalmaegi and Typhoon Morakot. The above conclusions are further strengthened in examining the correlation between the during-event precipitation and the LSI at severe landslides. The largest landslide scar in Typhoon Morakot (111.07 hectares) was located in the southwestern, which corresponds to the highest rainfall in Morakot but LSI indices range from low (0.06) to very high (0.30). In contrast, the largest landslide scar in Typhoon Kalmaegi (49.09 hectares) was distributed in the south near Mount Yushan, corresponding to a high LSI and low–moderate rainfall. This again shows that the precipitation factor in extreme events is extremely important, because it can cause landslides in all indices from low to very high.

## 4. Conclusions

Producing a disaster prediction map (landslide susceptibility map) is one of the important tasks that disaster researchers are interested in and need to carry out. They always try to develop statistical analysis models so that they can have the most accurate predictions of landslides in the future. Landslide researchers have constructed many models for landslide prediction by various methods, such as multivariate statistics, binary statistics, the AHP, neutral networks, etc. Especially, in using the AHP method for landslide interpretation, researchers often use the statistics of experts’ opinion and experience to establish weights or compare the importance for pairwise indicators (classes of each landslide factor).

The purpose of this paper is to present a new approach and via AHP modeling to generate landslide prediction maps (LSMs). In this study, a combination of the AHP, bivariate, and correlation statistics model was developed. Instead of using experts’ opinions to build a weighting matrix for the classes of factors, the bivariate analysis method (considering the landslide density of each class) is used to analyze this weight. In addition, correlation statistics were also used to evaluate the correlation coefficients of the landslide factors (*C*) by the AHP model. Finally, a multivariate analysis model incorporating the correlation coefficients of the landslide factors was applied to produce the LSM map.

To assess the accuracy of the above integration model, the LM data (the post-event landslide inventory maps, the during-event landslide inventory maps) associated with the events was used to evaluate the binary classification method, Kappa index, and success rate curves. The results showed that for normal events, such as Kalmaegi, LSM’s success rate is high (over 77% for post-event, and over 70% for during-event), whereas for extreme events, such as Morakot, it is less accurate (over 65% for post-event, and approximately 60% for during-event). This assessment proposed that further consideration of the rainfall data in landslide predictive models is needed in future work. Therefore, in the forthcoming work, we will continue to use this approach in developing a model for calculating landslide hazard indices, where the correlation coefficient between LSI (built on preparatory factors) and rainfall (a triggering factor) will be considered in the calculation model.

## Figures and Tables

**Figure 1 sensors-19-00505-f001:**
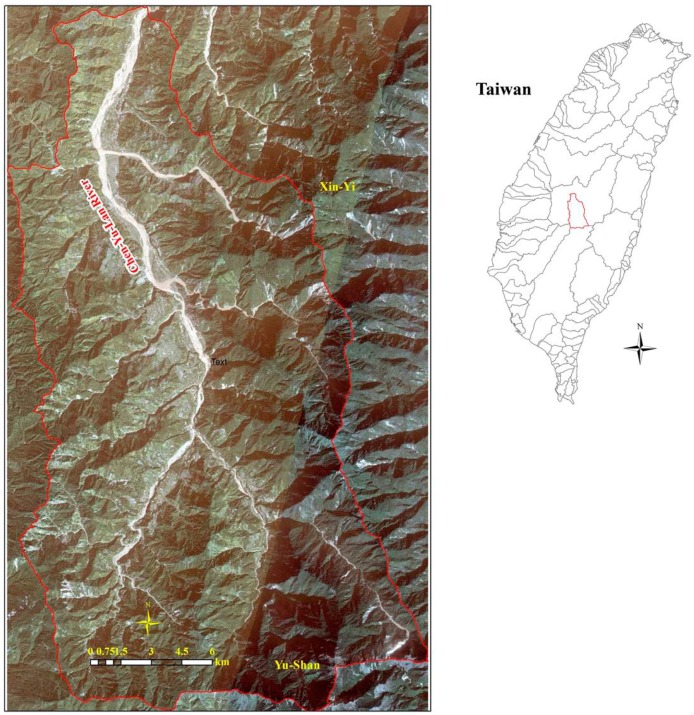
The study area.

**Figure 2 sensors-19-00505-f002:**
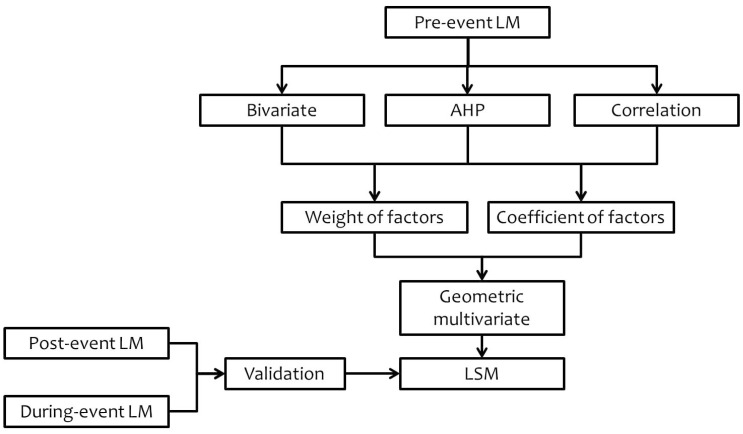
A flowchart of the research.

**Figure 3 sensors-19-00505-f003:**
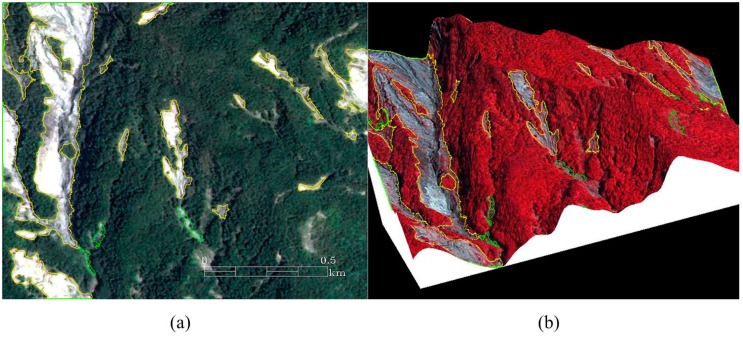
Detecting landslides from Formosat-2 satellite images by ELSADS: (**a**) 2D true-color composite image; and (**b**) Standard false-color composite image overlaid on the corresponding DEM.

**Figure 4 sensors-19-00505-f004:**
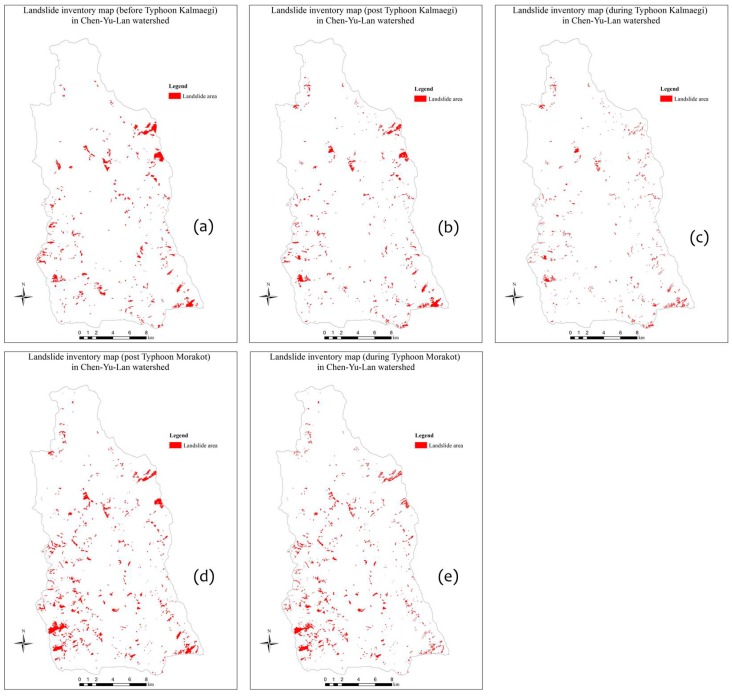
Types of LMs related to typhoons: (**a**) the pre-event Typhoon Kalmaegi LM, (**b**) the post-event Typhoon Kalmaegi LM, and (**c**) the during-event Typhoon Kalmaegi LM, (**d**) the post-event Typhoon Morakot LM, and (**e**) the during-event Typhoon Morakot LM.

**Figure 5 sensors-19-00505-f005:**
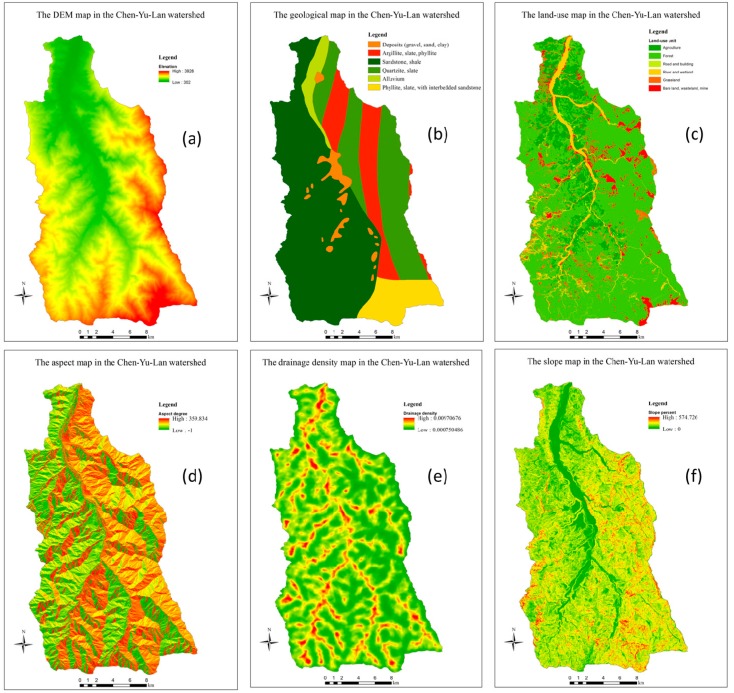
The material maps in the Chen-Yu-Lan watershed: (**a**) the DEM map; (**b**) the geological map; (**c**) the land-use map; (**d**) the aspect map; (**e**) the drainage density map; (**f**) the slope map.

**Figure 6 sensors-19-00505-f006:**
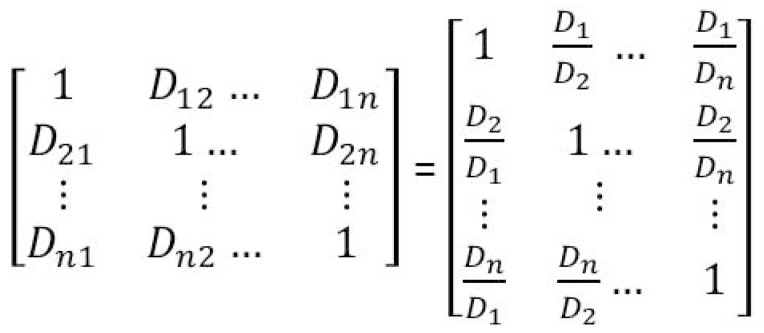
The matrix of pairwise comparison of classes in each factor.

**Figure 7 sensors-19-00505-f007:**
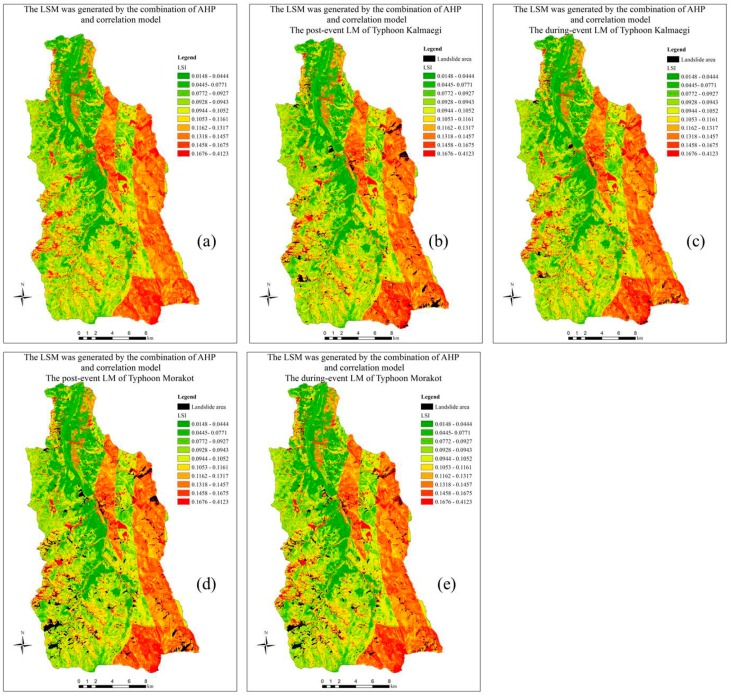
The LSM and related-event LM categories (**a**) The LSM was generated by the combination of AHP and correlation model; (**b**) The LSM overlapped with the post-event LM of Typhoon Kalmaegi; (**c**) The LSM overlapped with the during-event LM of Typhoon Kalmaegi; (**d**) The LSM overlapped with the post-event LM of Typhoon Morakot; and (**e**) The LSM overlapped with the during-event LM of Typhoon Morakot.

**Figure 8 sensors-19-00505-f008:**
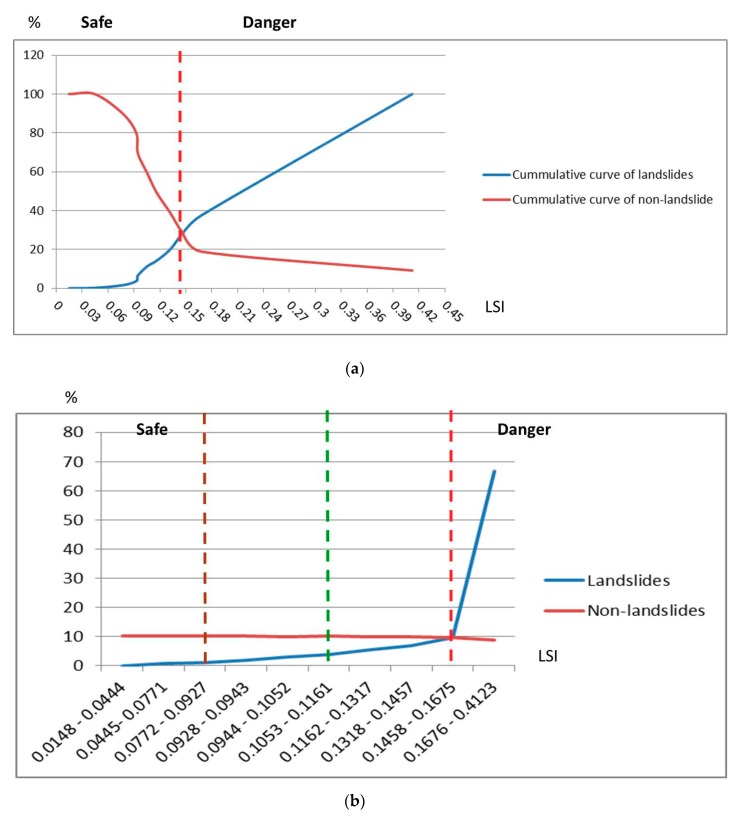
How to define thresholds of LSI. (**a**) Cumulative curves of landslides and non-landslides; (**b**) Distribution percentage curves of landslides and non-landslides.

**Figure 9 sensors-19-00505-f009:**
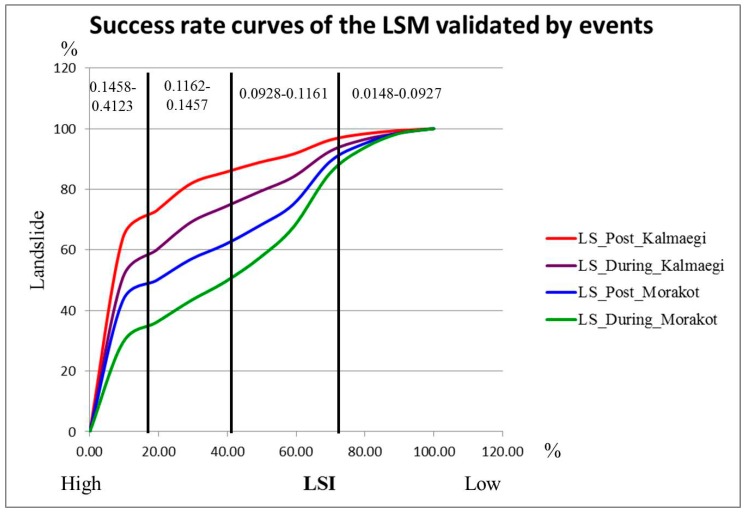
The success rate curves (landslide cumulative) of the LMs of events (Typhoon Kalmaegi and Typhoon Morakot).

**Table 1 sensors-19-00505-t001:** The table of random index (*RI*) [37].

*n*	1	2	3	4	5	6	7	8	9	10
*RI*	0	0	0.52	0.90	1.12	1.24	1.32	1.41	1.45	1.49

*n*: the matrix size (*n* × *n*).

**Table 2 sensors-19-00505-t002:** The variance–covariance matrix of factors.

	Factor 1	Factor 2	…	Factor *n*
Factor 1	*Var* _1_	*Co*var_12_	…	*Co*var_1*n*_
Factor 2	*Co*var_21_	*Var* _2_	…	*Co*var_2*n*_
:	:	:	…	:
Factor *n*	*Co*var_*n*1_	*Co*var_*n*2_	…	*Var_n_*

**Table 3 sensors-19-00505-t003:** The correlation matrix of landslide factors.

	Factor 1	Factor 2	…	Factor *n*
Factor 1	1	*r* _12_	*…*	*r* _*n*1_
Factor 2	*r* _21_	1	*…*	*r* _*n*2_
:	:	:	*…*	:
Factor *n*	*r* _*n*1_	*r* _*n*2_	*…*	1

**Table 4 sensors-19-00505-t004:** Calculating the weight of factors by AHP.

		(1)	(2)	(3)	(4)	(5)	(6)	(7)	(8)	Weight (W)
Slope ^1^ (SL)	(1)	1	0.231	0.058	0.039	0.033	0.009	0.004		0.002
	(2)	4.326	1	0.253	0.170	0.142	0.040	0.017		0.010
	(3)	17.134	3.960	1	0.671	0.564	0.158	0.066		0.039
	(4)	25.158	5.898	1.489	1	0.840	0.236	0.100		0.057
	(5)	30.365	7.019	1.772	1.190	1	0.280	0.118		0.068
	(6)	108.313	25.036	6.322	4.245	3.567	1	0.420		0.243
	(7)	257.971	59.628	15.056	10.109	8.496	2.382	1		0.580
Drainage density ^2^ (DD)	(1)	1	1.176	1.092	0.964	1.080	1.665			0.188
	(2)	0.850	1	0.938	0.819	0.918	1.415			0.159
	(3)	0.916	1.077	1	0.883	0.989	1.525			0.172
	(4)	1.038	1.221	1.133	1	1.121	1.728			0.195
	(5)	0.926	1.089	1.011	0.892	1	1.542			0.174
	(6)	0.601	0.706	0.656	0.579	0.649	1			0.113
Lithology ^3^ (GM)	(1)	1	0.393	0.329	0.130	0.952	0.115			0.042
	(2)	2.545	1	0.838	0.331	2.422	0.293			0.106
	(3)	3.036	1.193	1	0.395	2.890	0.349			0.126
	(4)	7.687	3.020	2.532	1	7.136	0.884			0.320
	(5)	1.051	0.413	0.346	0.137	1	1.121			0.044
	(6)	8.694	3.416	2.863	1.131	8.274	1			0.362
Land-use ^4^ (LU)	(1)	1	0.066	0.487	0.008	0.084	0.003			0.002
	(2)	15.261	1	7.437	0.124	1.289	0.052			0.034
	(3)	2.052	0.134	1	0.017	0.173	0.007			0.005
	(4)	123.081	8.065	59.982	1	10.399	0.419			0.275
	(5)	11.836	0.776	5.768	0.096	1	0.040			0.026
	(6)	294.059	19.269	143.307	2.389	24.845	1			0.657
Aspect ^5^ (AS)	(1)	1	0.982	0.947	0.933	0.478	0.560	0.526	0.916	0.111
	(2)	1.018	1	0.964	0.950	0.486	0.571	0.537	0.933	0.113
	(3)	1.056	1.037	1	0.986	0.504	0.592	0.556	0.967	0.117
	(4)	1.071	1.052	1.014	1	0.511	0.600	0.564	0.981	0.119
	(5)	2.094	2.057	1.983	1.954	1	1.173	1.102	1.918	0.233
	(6)	1.784	1.753	1.690	1.666	0.852	1	0.939	1.635	0.198
	(7)	1.901	1.867	1.800	1.774	0.908	1.065	1	1.741	0.211
	(8)	1.092	1.072	1.034	1.019	0.521	0.612	0.574	1	0.121

Slope ^1^: below 5% (1); 5–15% (2); 15–30% (3); 30–40% (4); 40–55% (5); 55–100% (6); greater than 100% (7). Drainage density ^2^: less than 0.0015 m/m^2^ (1); 0.0015–0.002 m/m^2^ (2); 0.002–0.0025 m/m^2^ (3); 0.0025–0.003 m/m^2^ (4); 0.003–0.004 m/m^2^ (5); greater than 0.004 m/m^2^ (6). Lithology ^3^: deposit (1); argillite, slate, phyllite (2); sandstone, shale, basaltic rock (3); quartzite, slate, coaly shale (4); alluvium (5); phyllite, slate, with interbedded sandstone (6). Land-use ^4^: agriculture (1); forest (2); road and building (3); grassland (4); river and wetland (5); bare land, wasteland, mines (6). Aspect ^5^: North (1); Northeast (2); East (3); Southeast (4); South (5); Southwest (6); West (7); Northwest (8).

**Table 5 sensors-19-00505-t005:** The *CR*, *RI*, and *CI* of calculating factors’ weights by the AHP.

Factor	SL	AS	DD	GM	LU
*CR*	2.24 × 10^−16^	0	0	0	2.86 × 10^−16^
*RI*	1.32	1.41	1.24	1.24	1.24
*CI*	2.96 × 10^−16^	0	0	0	3.55 × 10^−16^

**Table 6 sensors-19-00505-t006:** The matrix of landslide factors’ coefficients.

Factor	SL	AS	DD	GM	LU	Coefficient (*C*)
SL	1	0.000401	0.000038	0.000871	0.00046	0.1449
AS	0.000401	1	0.000029	0.001643	0.002203	0.2132
DD	0.000038	0.000029	1	0.000041	0.000033	0.0275
GM	0.000871	0.001643	0.000041	1	0.005867	0.3245
LU	0.00046	0.002203	0.000033	0.005867	1	0.2900

*CI* = (−0.99), *RI* = 1.12, and *CR* = (−0.89); In this case, the consistency ratio is less than 0.1 (acceptable).

**Table 7 sensors-19-00505-t007:** A statistical table of validating results (Typhoon Kalmaegi and Typhoon Morakot).

The LMs		*TP* (%)	*TN* (%)	*FP* (%)	*FN* (%)	Accuracy (%)	Error (%)	*P_e_*	*P* _0_	*K*
**Typhoon Kalmaegi**	Post-event	73.46	82.44	26.54	17.56	77.35	22.65	0.5	0.779	0.559
	During-event	59.08	82.11	40.92	17.89	70.26	29.74	0.5	0.706	0.412
**Typhoon Morakot**	Post-event	49.86	81.42	50.14	18.58	65.64	34.36	0.5	0.656	0.313
	During-event	36.04	80.93	63.96	19.07	58.49	41.51	0.5	0.585	0.170

**Table 8 sensors-19-00505-t008:** Statistics of landslides which occurred in the events (Typhoon Kalmaegi and Typhoon Morakot).

LSI Level	LSI	A (%)	Post Typhoon Kalmaegi		During Typhoon Kalmaegi		Post Typhoon Morakot		During Typhoon Morakot	
			LS (%)	NLS (%)	LS (%)	NLS (%)	LS (%)	NLS (%)	LS (%)	NLS (%)
Low	0.0148–0.0444	10.04	0.60	10.16	1.43	10.08	1.40	10.25	1.54	10.16
	0.0445–0.0771	10.24	1.13	10.36	2.19	10.28	3.58	10.40	4.84	10.32
	0.0772–0.0927	9.97	2.04	10.07	3.89	10.00	5.96	10.06	8.41	9.99
Moderate	0.0928–0.0943	10.59	4.62	10.67	8.41	10.60	14.04	10.51	17.75	10.48
	0.0944–0.1052	9.87	2.75	9.96	4.88	9.89	7.08	9.93	10.15	9.86
	0.1053–0.1161	10.07	3.29	10.16	5.03	10.09	6.20	10.16	7.94	10.10
High	0.1162–0.1317	9.87	3.73	9.95	5.07	9.89	4.84	9.99	6.13	9.92
	0.1318–0.1457	10.05	8.98	10.06	9.22	10.05	7.03	10.12	7.20	10.09
Very high	0.1458–0.1675	9.66	8.79	9.67	8.89	9.67	6.48	9.74	6.58	9.71
	0.1676–0.4123	9.65	64.06	8.93	50.99	9.45	43.38	8.84	29.47	9.36

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
