# Peer review of "A New Approach Using AHP to Generate Landslide Susceptibility Maps in the Chen-Yu-Lan Watershed, Taiwan"

_sensors, 2019, doi:10.3390/s19030505_

Round 1

Reviewer 1 Report

This paper is very well prepared and written. The data sets and methodology developed and used are described in detail. The approach of quantifying risks associated with landslides based on a combination of AHP and statistical analysis instead of expert opinions is unique. The authors evaluated thoroughly and successfully the proposed methodology based on historical data for Taiwan. I am confident that other regions and countries that are vulnerable to similar landslide risks will find the methodology to be very useful to them, and most probably will adopt. I do hope the authors are willing to share details of their methodologies with such interested users. It might be worthwhile to have short sentence in the paper expressing such availability to interested users. I recommend publication.

Author Response

Responses to Reviewer 1 comments

This paper is very well prepared and written. The data sets and methodology developed and used are described in detail. The approach of quantifying risks associated with landslides based on a combination of AHP and statistical analysis instead of expert opinions is unique. The authors evaluated thoroughly and successfully the proposed methodology based on historical data for Taiwan. I am confident that other regions and countries that are vulnerable to similar landslide risks will find the methodology to be very useful to them, and most probably will adopt. I do hope the authors are willing to share details of their methodologies with such interested users. It might be worthwhile to have short sentence in the paper expressing such availability to interested users. I recommend publication.

- Thank you so much for your comments. Based on reviewers' comments, a few sentences in the manuscript have been edited and added in some places to clarify the points. All of them have been highlighted in the article. Please check it.

Reviewer 2 Report

 1. P2L55, “more than 10 hurricanes hit Taiwan annually, and most landslides are 56 caused by typhoons or hurricanes”
Is “hurricane” correct? Should it be “typhoon”?

2. P2L83, “In most previous AHP studies”
Add references.

3. P3L143, “30,00 mm”
Is this “3,000 mm”?

4. P3L144, “These typhoons represent normal and extreme events”
Clarify the definitions of “normal” and “extreme” in this article.

5. Figure 1
Add the scale.

6. Figure 2
Add the scale.

7. P6L186, “wans”
What is this?

8. Table 4
Clarify the unit of “Drainage density”

9. P12L303, “The LSM showed that … (Figure 7b-e).”
This explanation and the description in P15L379, “The largest landslide … but moderate LSI.”, are not consistent. Please clarify.

10. P12L310, “based on the variation of two above curves … (the dotted brown and green lines).”
It is difficult to understand how these two sub-thresholds are defined. Please add explanation.

11. P12L312, “low (index: 0.0148-0.0927), medium (0.0928-0.1161)”
The value “0.0927” and “0.0938” do not coincide with the value indicated by the brown dotted line in Figure 8. Please clarify.

12. Figure 8
The value of “Cumulative curve of non-landslides at the point of LSI 0.4123 is not 0% although this value seems to be the maximum in the study area. Please clarify.

13. P14L358, “To increase interpreting agreement … into the computation model”
Is this “the computation model” the phase one model or the phase two model (P2L58). Please clarify.

14. P15L382, “This again shows that … for extreme events.”
Rainfall by Typhoon Kalmaegi is big enough even those it may not be classified as extreme event, and the events with less rainfall are not examined in this article. Is it true that the precipitation factor can be neglected for normal events?

Author Response

Responses to Reviewer 2 comments

Based on the questions and comments you have suggested for us, we would like to summarize the following points:

1. P2L55, “more than 10 hurricanes hit Taiwan annually, and most landslides are 56 caused by typhoons or hurricanes”

Is “hurricane” correct? Should it be “typhoon”?

- This sentence has been revised. ”Typhoon” has been erased from the sentence. Please check P2L55-56.

2. P2L83, “In most previous AHP studies”

Add references.

- References have been added in this sentence. Please check P2L83-84.

3. P3L143, “30,00 mm”

Is this “3,000 mm”?

- This information has been corrected from “30,00mm” into “3,000mm”. Please check P3L144.

4. P3L144, “These typhoons represent normal and extreme events”

Clarify the definitions of “normal” and “extreme” in this article.

- An explanation of why we use "normal" and "extreme" to call Typhoon Kalmaegi and Typhoon Morakot was added. Please check P4L145-152.

5. Figure 1

Add the scale.

- The scale has been added on Figure 1. Please check P4L155

6. Figure 2

Add the scale.

- The scale has been added on Figure 1. Please check P5L171

7. P6L186, “wans”

What is this?

- This is a typing mistake. This word “wans” has been deleted. Please check P6L193.

8. Table 4

Clarify the unit of “Drainage density”

- The unit of “Drainage density” has been added to the “table footer” section of Table4. Please check P11L293-294.

9. P12L303, “The LSM showed that … (Figure 7b-e).”

This explanation and the description in P15L379, “The largest landslide … but moderate LSI.”, are not consistent. Please clarify.

- This sentence has been revised and improved. Please read P16L412-417.

10. P12L310, “based on the variation of two above curves … (the dotted brown and green lines).”

It is difficult to understand how these two sub-thresholds are defined. Please add explanation.

- An explanation of why and how two sub-thresholds were defined was added. Please check P12L315-329.

11. P12L312, “low (index: 0.0148-0.0927), medium (0.0928-0.1161)”

The value “0.0927” and “0.0938” do not coincide with the value indicated by the brown dotted line in Figure 8. Please clarify.

- An explanation of how to design thresholds has been added and revised, and Figure 8 has been corrected. Please check P12L315-329 and P14L352.

12. Figure 8

The value of “Cumulative curve of non-landslides at the point of LSI 0.4123 is not 0% although this value seems to be the maximum in the study area. Please clarify.

- An explanation of the above case has been added in manuscript. Please check P13L338-345.

13. P14L358, “To increase interpreting agreement … into the computation model”

Is this “the computation model” the phase one model or the phase two model (P2L58). Please clarify.

- This sentence has been rewritten and supplemented to make it clear. Please read P15L384-392.

14. P15L382, “This again shows that … for extreme events.”

Rainfall by Typhoon Kalmaegi is big enough even those it may not be classified as extreme event, and the events with less rainfall are not examined in this article. Is it true that the precipitation factor can be neglected for normal events?

- An explanation of why we proposed that Typhoon Kalmaegi is a “normal event”. Please check P4L145-152.

- To clarify this point, we edited some places and rewritten this sentence. Please check P15L384-392 and P16L412-417.